# Relational VAE: A Continuous Latent Variable Model for Graph Structured Data

## Abstract

Graph Networks (GNs) enable the fusion of prior knowledge and relational reasoning with flexible function approximations. In this work, a general GN-based model is proposed which takes full advantage of the relational modeling capabilities of GNs and extends these to probabilistic modeling with Variational Bayes (VB). To that end, we combine complementary pre-existing approaches on VB for graph data and propose an approach that relies on graph-structured latent and conditioning variables. It is demonstrated that Neural Processes can also be viewed through the lens of the proposed model. We show applications on the problem of structured probability density modeling for simulated and real wind farm monitoring data, as well as on the meta-learning of simulated Gaussian Process data. We release the source code, along with the simulated datasets.

## 1 Introduction

Graph Neural Networks (GNNs) [1, 2] have been established as an effective tool for representation learning on graph structured data. Graph structured data are routinely employed to represent entities and relations among them. The present work focuses in representation of uncertainty and generative modeling for attributed directed graph data with continuous attributes. The initiating motivation for this work is the ubiquity of noisy structured data and systems with stochastic or partially observable interactions of industrial relevance (e.g. wind farms and urban transportation networks).

In the context considered herein, modeled entities (*nodes*) and modeled relations (*edges*) may feature a *state*, which may not be fully observed and/or stochastic. The same may also holds for global (*graph*) attributes. At the same time, nodes and relations may possess a dynamic partially observed state, which we may infer directly from data. Both the node states and edge states are not fully observed and non-deterministic, which amply motivates probabilistic extensions of graph networks. In essence, this work proposes a method that 1) exploits the relational structure of data and 2) allows for learning flexible distributions over entity and relation attributes. Several partially overlapping approaches for this problem exist. A short review of such prior approaches is offered in section 3. Modeling entities and relations has been shown empirically to allow for stronger generalization [3, 4, 5] in novel settings. The main contribution of this work is to propose an approach to transfer the potent combinatorial generalization and modeling capabilities of GNNs to the problem of modeling conditional distributions of structured data.

## 2 Methods

**Attributed graphs** Following [2], global attribute augmented graphs are denoted by $G = (\mathcal{V}, \mathcal{E}, \mathbf{u})$ where $\mathcal{V} : \{\mathbf{v}_i\}_{i=1:N^v}$ with $\mathbf{v}_i \in \mathbb{R}^{d^v}$ denoting the nodes (vertices) of the graph,

Submitted to 35th Conference on Neural Information Processing Systems (NeurIPS 2021). Do not distribute.

34   $\mathcal{E} : \{(\mathbf{e}_k, s_k, r_k)\}_{k=1:N^e}$ designating the set of edges, with edge attributes $\mathbf{e}_k \in \mathbb{R}^{d^e}$, $s_k, r_i \in \mathbb{N}^1$

35   denote the head (sender) and tail (receiver) nodes of the modeled relation, while $\mathbf{u} \in \mathbb{R}^{d^u}$ is the

36   global attribute.

37   **Graph Networks (GN)**   (or GraphNets) are composite functions that receive and return attributed

38   graphs. The full GN block consists of an edge update, a node update and a global update block. Each

39   block contains a corresponding function $\phi^e, \phi^v, \phi^u$. The edge update function uses edge, node and

40   global data. The edge block is followed by an aggregation step $\rho^{e \to n}$, where edge messages are

41   accumulated according to a permutation invariant function, e.g. a mean function. The node update

42   uses (optionally) the global state, the aggregated edge state and the current node state. Finally, a global

43   block aggregates with permutation invariant functions the edge and node properties ($\rho^{e \to u}, \rho^{v \to u}$),

44   and optionally uses the global state for updating the global variable state. Different parts of the full

45   GN computation may be omitted. Several Graph Neural Network architectures can be cast as special

46   cases of GNs by omitting certain features or by special choices of the different functions involved

47   [2]. In what follows, when referring to GNNs, the most general and expressive GN layer is implied

48   except otherwise specified.

49   In the proposed model, entities (nodes), relations (edges) and global attributes contain both determin-

50   istic and stochastic variables. These variables in turn, may be observable or not directly observable.

51   Both observable and unobservable attributes may be deterministic or stochastic (static or evolving).

52   In what follows, a part of the observable quantities is referred to as *conditioning* or *context*. The node,

53   edge and global observable quantities are denoted as $\mathbf{v}^h, \mathbf{e}^h, \mathbf{u}^h$ where $h$ signifies that a variable

54   corresponds to conditioning. Conditioning variables may either correspond to conditioning with

55   known dynamic quantities or static quantities. Common instantiations of such conditioning are

56   positional encoding for vertices, relative position for edges between vertices and time of day as

57   a global variable. The node, edge and global variables that correspond to the rest of the states

58   (stochastic, evolving, unobserved) are denoted by $\mathbf{v}^d, \mathbf{e}^d, \mathbf{u}^d$. In essence, the conditioning attributes

59   can be used to create a *conditioning graph variable* $G_h = (\mathcal{V}_h, \mathcal{E}_h, \mathbf{u}_h)$ and a *state graph variable*

60   $G_x = (\mathcal{V}_x, \mathcal{E}_x, \mathbf{u}_x)$. The full graph state, is denoted by $G_d = (\mathcal{V}_x \cup \mathcal{V}_h, \mathcal{E}_x \cup \mathcal{E}_h, \mathbf{u}_x \cup \mathbf{u}_h)$ where

61   $\cup$ denotes set union. Since part of the node, edge and global attributes may be stochastic, a graph

62   structured latent variable $G_z = (\mathcal{V}_z, \mathcal{E}_z, \mathbf{u}_z)$ is assumed. The graph structure may also be determined

63   through the edge variables as in [6], but we restrict our model to a pre-determined graph structure

64   in this work. The following model is proposed for the joint distribution of the graph structured

65   observations

$$p(G_x; G_h) = \int p(G_x | G_z; G_h) p(G_z; G_h) dG_z \tag{1}$$

66   where $p(G_z; G_h) = p(\mathcal{V}_z; \mathcal{V}_h) p(\mathcal{E}_z; \mathcal{E}_h) p(\mathbf{u}_z; \mathbf{u}_h)$ is the distribution of the latent variables given $G_h$.

67   A prior distribution conditioned on $G_h$ is assumed for the latent variable, which is further factorized

68   along each edge and node latent separately, i.e.,

$$p(G_z; G_h) = p^{(\mathcal{V})}(\mathcal{V}_z; \mathcal{V}_h) p^{(\mathcal{E})}(\mathcal{E}_z; \mathcal{E}_h) p^{(\mathbf{u})}(\mathbf{u}_h; \mathbf{u}_z) \tag{2}$$

$$= \prod_{i=1}^{N^v} p(\mathbf{v}_i^z; \mathbf{v}_i^h) \cdot \prod_{k=1}^{N^e} p(\mathbf{e}_k^z; \mathbf{e}_k^h) \cdot p(\mathbf{u}^z; \mathbf{u}^h). \tag{3}$$

69   An approximate posterior (i.e., *recognition model*) is assumed for $G_z$ as $q_\phi(G_z | G_x; G_h)$ together

70   with a generative model for $G_x$, $p_\theta(G_x | G_z; G_h)$. In correspondence with the Variational Autoen-

71   coder (VAE) [7], we seek to learn the generative model parameters $\boldsymbol{\theta}$ and inference model param-

72   eters $\phi$ simultaneously. Assuming independent identically distributed (i.i.d.) graph observations

73   $\{G_x^{(1)}, \dots G_x^{(i)}\}$, the Evidence Lower Bound (ELBO) for the marginal log-likelihood reads

$$\mathcal{L}(\boldsymbol{\theta}, \boldsymbol{\phi}; G_x^{(i)}, G_h^{(i)}) = \mathbb{E}_{q_\theta(G_z | G_x^{(i)}; G_h^{(i)})} \big[ \log p_\theta(G_x^{(i)} | G_z; G_h^{(i)}) \big]$$
$$- D_{KL}(q_\phi(G_z | G_x^{(i)}; G_h^{(i)}) || p_\theta(G_z; G_h^{(i)})) \tag{4}$$

74   We seek to perform fast and scalable approximate inference over the $G_z$ graph variable and at the

75   same time take advantage of the *relational structure* in the data. A particularly convenient choice

76   for parametrizing $G^z$ is to assume a parametric distribution over edges, nodes and globals. A GN is

proposed for inferring the parameters. For a graph structured observation observation $G_x$, we write

$$\mathcal{V}^z \sim q_{q\phi}^{(\mathcal{V})}(G_z|G_x;G_h) = \mathcal{N}(f_{q\phi}^{\mu(\mathcal{V})}(G_x;G_h), f_{q\phi}^{\sigma^2(\mathcal{V})}(G_x;G_h)) \tag{5}$$

$$\mathcal{E}^z \sim q_{q\phi}^{(\mathcal{E})}(G_z|G_x;G_h) = \mathcal{N}(f_{q\phi}^{\mu(\mathcal{E})}(G_x;G_h), f_{q\phi}^{\sigma^2(\mathcal{E})}(G_x;G_h)) \tag{6}$$

$$\mathbf{u}_z \sim q_{q\phi}^{(\mathbf{u})}(G_z|G_x;G_h) = \mathcal{N}(f_{q\phi}^{\mu(\mathbf{u})}(G_x;G_h), f_{q\phi}^{\sigma^2(\mathbf{u})}(G_x;G_h)). \tag{7}$$

The functions $f_\cdot^{\mu(\cdot)}$ and $f_\cdot^{\sigma^2(\cdot)}$ are implemented by a GN to allow for taking into account in a general manner relational information while inferring over $\mathcal{V}_z, \mathcal{E}_z$ and $\mathbf{u}_z$. In practice a shared, single GN, $f_{q\phi}(\cdot)$ is used. The parametrization for vertices, edges and global variables are the corresponding states of the GN at the final message passing step. In a similar manner, a GN generator network, $g_{p\theta}(\cdot)$, is used for $p_\theta$. Since the prior and posterior are factorized over nodes, edges and the global variable of each graph datapoint, the ELBO is split accordingly as

$$\begin{aligned}
\mathcal{L}(\boldsymbol{\theta}, \boldsymbol{\phi}; G_x^{(i)}, G_h^{(i)}) =& \mathbb{E}_{q_\theta(G_z|G_x^{(i)};G_h^{(i)})}\big[\log p_\theta(G_x^{(i)}|G_z;G_h^{(i)})\big] \\
& - \beta_\mathcal{V} D_{KL}(q_\phi^{(\mathcal{V})}(G_z|G_x^{(i)};G_h^{(i)})||p_\theta^{(\mathcal{V})}(G_z;G_h^{(i)})) \\
& - \beta_\mathcal{E} D_{KL}(q_\phi^{(\mathcal{E})}(G_z|G_x^{(i)};G_h^{(i)})||p_\theta^{(\mathcal{E})}(G_z;G_h^{(i)})) \\
& - \beta_\mathbf{u} D_{KL}(q_\phi^{(\mathbf{u})}(G_z|G_x^{(i)};G_h^{(i)})||p_\theta^{(\mathbf{u})}(G_z;G_h^{(i)}))
\end{aligned} \tag{8}$$

where $\beta_\mathcal{V}, \beta_\mathcal{E}, \beta_\mathbf{u}$ can be used for controlling disentanglement as in $\beta$–VAE [8] or the rate-distortion characteristics of the model [9] or for preventing posterior collapse and aiding training through *KL-annealing* [10, 11]. In a similar manner to VAEs, the approach to representing distributions over graph data with a distribution that factorizes over $\mathcal{V}, \mathcal{E}$ and $\mathbf{u}$ allows for defining alternative evidence lower bounds for variational Bayes. Note that the distribution does not need to be factorized along the elements of the latent vector. This allows straight-forward extensions using more flexible distributions [12]. A generative model based on normalizing flows that uses shift-scale transformations [13] has already been proposed in [14] for graph generation. The Relational VAE (RVAE) model proposed can be extended as a hierarchical VAE [15] yielding a model akin to Doubly Stochastic Variational Neural Process (DSNPV) [16], which uses global and node variables. Finally, Neural Processes [17, 18] (NP) and other graph encoder-decoder models [6, 19, 20, 21, 22] are closely related to the proposed model.

## 3 Related work

**GNN Encoder-decoder models**   In Neural Relational Inference (NRI) [6] discrete edge latent variables are inferred from node representations and a re-parametrized $Gumbel - Softmax$ distribution is used[23, 24]. A coarse representation of the computational graphs of NRI, NPs and the RVAE is shown in Figure 1. In [19] graphs are modeled from global continuous latent variables, which are subsequently used for graph generation through an adjacency matrix. In GraphVAE [20] the global variable together with a graph-structured conditioning variable is used for generation. In *Graphite* [21] a latent variable for each node is inferred from the encoder, while the edge variables (i.e., symmetric adjacency matrix) is inferred through efficient iterated message passing. Similarly, the VariationalGAE[22] uses a separate latent variable for every node and a graph convolutional encoder. Several of the aforementioned works take advantage of recent advances in low-variance gradient estimates for distributions over latent variables, as in Variational Autoencoders (VAEs) via the reparametrization trick [7, 25]. The overlapping traits of the aforementioned are the treatment of edge, node and global variables. In Table 1 a summary of the relational modeling capabilities of various graph encoder-decoder models is offered. Note that the table highlights only the relevant parts to this work together with several important and influential design choices for graph representation learning were not touched upon. For instance, the graph convolutional models of some of the aforementioned works offer the important advantage of scalability and small computation cost.

In this work, the above mentioned approaches, are generalized and unified in the proposed Relational Variational Autoencoder (RVAE) model. Note that it is not difficult to yield explicit graph connectivity in RVAE as in NRI [6] since the type and existence of a connection can be seen as a categorical variable. See also Figure 1 (b), where a sketch of NRI is offered. Inferring graph connectivity or

generating graphs, however, falls out of the scope of this work. In RVAE the focus is generative modeling of graph structured data with an apriori known connectivity, with attributed nodes and edges, which optionally may include a global attribute that influences both entities and relations.

Table 1: Features of different related Bayesian graph network encoder-decoder models (see also Figure 1). For the NP models that contain a latent variable, it is straightforward to combine a deterministic global encoder for the context inputs at test time [26]. The attributes with subscript $z$ denote that the model performs optimization using an ELBO objective. The attributes with superscript $h$ denote whether the models may facilitate deterministic conditioning for the corresponding graph attribute at test time.

| | Latent | | | Conditioning | | | |
| Name | $\mathcal{V}_z$ | $\mathcal{E}_z$ | $\mathbf{u}_z$ | $\mathcal{V}_h$ | $\mathcal{E}_h$ | $\mathbf{u}_h$ | Architecture notes |
|---|---|---|---|---|---|---|---|
| CNP [17] | – | – | – | ✓ | – | ✓ | DeepSet encoder, GN node block |
| AttCNP [27] | – | – | – | ✓ | ✓ | (✓) | Attention encoder/decoder |
| | | | | | | | Decoder edge cond. through cross-attention |
| ConvCNP [28] | – | – | – | ✓ | ✓ | (✓) | SetConv encoder |
| NP [18] | – | – | ✓ | ✓ | – | (✓) | DeepSet encoder |
| GraphVAE [20] | – | – | ✓ | ✓ | ✓ | ✓ | Graph conv. encoder |
| VariationalGAE [22] | ✓ | – | – | – | – | – | Graph convolutions |
| Graphite [21] | ✓ | – | – | ✓ | – | – | Iterative decoder |
| NRI [6] | – | ✓ | – | ✓ | – | – | MP encoder/decoder |
| MPNP [29] | – | – | ✓ | ✓ | ✓ | (✓) | MP encoder/decoder |
| DSVNP [16] | ✓ | – | ✓ | ✓ | – | (✓) | $\mathcal{V}^z \sim p(\mathcal{V}^z|\mathbf{u}^z, \mathcal{V}^*, \mathcal{V}^{h*})$ |
| RVAE (this work) | ✓ | ✓ | ✓ | ✓ | ✓ | ✓ | MP encoder/decoder |

**Neural processes** In Neural Processes (NP)[17, 18], we consider a set of mappings $F : \mathcal{X} \to \mathcal{Y}$ where $\mathcal{X} : \{x_i\}$, $x_i \in \mathbb{R}^{N_x}$, $\mathcal{Y} : \{y_i\}$, $y_i \in \mathbb{R}^{N_y}$. A particular draw of a function $f \sim F$, is modeled as $f(x_i) = g_\theta(x_i, z)$ where $z \sim p(z)$ is a high dimensional random vector (e.g. a standard normal) and $g_\theta$ is a neural network and $\theta$ denotes the parameters of $g$. Given a set of $n_m$ input-output observations $\mathcal{D} : \{(x_{1:n_m}, y_{1:n_m})_{f_m}\}$ from $m$ different realizations of $f$ (potentially different in number), we want to learn a distribution over $z \sim p(z|\mathcal{D})$. Under the NP approximation, assuming observation noise $y_i \sim \mathcal{N}(g_\theta(x_i, z), \sigma^2)$, the distribution of $y$ is defined as

$$p(z, y_{1:n}|x_{1:n}) = p(z) \prod_{j=1}^{n} \mathcal{N}(y_i|g(x_i, z), \sigma^2). \tag{9}$$

In practice, the input-output observation cases $\mathcal{D}$, are split as $\mathcal{D}^{C \cup T} = \mathcal{D}^C \cup \mathcal{D}^T$, where $C$ denotes a set of points with observations in $\mathcal{X}$ and $\mathcal{Y}$ and $T$ denotes a set of points where we only observe $\mathcal{X}$ (i.e., the inputs). This can be cast as a conditional generative model for $p(y_T|x_T, x_C, y_C) = p_\theta(y_T|x_T, z)p(z|x_C, y_C)$, where the conditioning is the fully observed context pairs. The ELBO used for optimization is

$$\log p(y_T|x_T, x_C, y_C) \geq \mathbb{E}_{q_\phi(z|\mathcal{D}^{C \cup T})} \Big[ \sum_{i \in T} \log p_\theta(y_i|z, x_i) + \log \frac{q(z|\mathcal{D}^C)}{q(z|\mathcal{D}^{C \cup T})} \Big]$$

$$\mathbb{E}_{q_\phi(z|\mathcal{D}^{C \cup T})} \Big[ \sum_{i \in T} \log p_\theta(y_i|z, x_i) \Big] - D_{KL}(q_\phi(z|\mathcal{D}^{C \cup T})||q_\phi(z|\mathcal{D}^C)) \tag{10}$$

Note that the above variational objective has an intuitive interpretation, as a reconstruction loss (first part) and a Kullback-Leibler divergence between the approximate posterior distributions predicted when using both $C \cup T$ and when using only $C$ (the context set). In [30] a similar loss function was proposed with the motivation of training VAEs that can be used with arbitrary conditioning masks. By considering the set of observations as nodes in a disconnected graph, (i.e., $\mathcal{V} : \{\mathbf{v}_i|(x_i, y_i)\}_{i=1:N^v}$) and training while masking the context output nodes $y_C$, the same objective is retrieved. Therefore, following the nomenclature of [2], we can instantiate a NP from the proposed model, by using arbitrary conditioning as described in [30], a DeepSet [31] as an encoder and only a node-block as a decoder as shown in Figure 1.

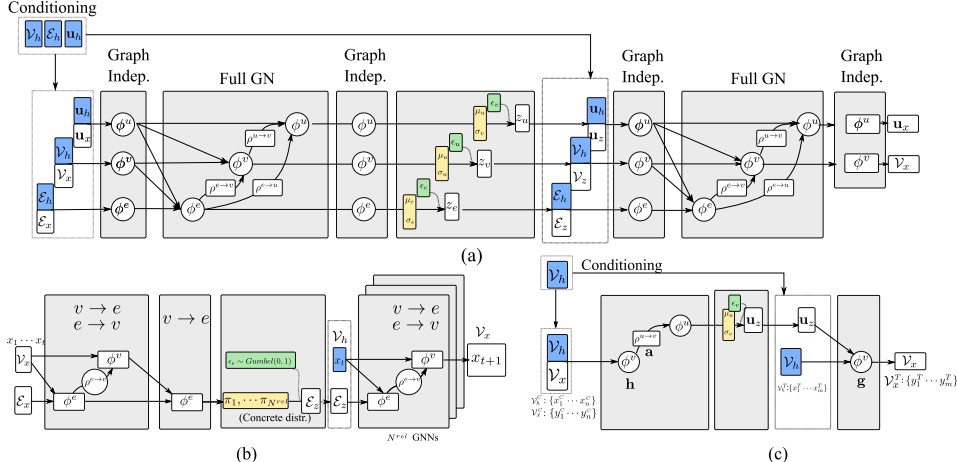

Figure 1: **(a)** Proposed architecture with a single message passing step in the encoder and decoder **(b)** the Neural Relational Inference model of [6]. **(c)** The Neural Process model [18]. For direct correspondence between the present work and [6] and [18] the notations of the other works are included in the figure (e.g. $\rho^{\mathbf{v}\to\mathbf{u}} = \mathbf{a}$ in the Neural Process model).

The NP framework has been extended to take advantage of special inductive biases, such as the *relation* of observation and target nodes in Attentive Conditional Neural Processes (AttNP) [27] or the translation equivariance in Convolutional Conditional Neural Processes (ConvCNP) [28]. More recently, relational inductive biases were employed in Message Passing Neural Processes (MPNP) [29]. The aforementioned models, feature a global latent variable $\mathbf{u}_z$ which is inferred from the context points and parametrizes the distribution over functions. With the exception of MPMP, the aforementioned works target non-relational data. Nevertheless, MPMPs does not directly implement edge-bound uncertainty or edge-level conditioning, which is the most pronounced difference to RVAE. Similar to this work, in DSNPV [16] a NP that allows for both node and global latent variables was proposed, which in addition, employs a hierarchical VAE [15]. The motivation of DSNPV is to include node-context information, which in the conditional RVAE is also supported by design through $\mathcal{V}_h$. RVAE attempts to merge the complementary strengths of the aforementioned models in representation of uncertainty, with a focus towards modeling graph structured continuous data. Finally, in contrast to Functional Neural Processes [32] we do not deal with inferring a graph of dependencies among latent variables, yet hierarchical RVAE adaptations may also manage such tasks.

**Graph Gaussian processes**  Sharing the motivation of this work, i.e., taking advantage of relational information and learning joint distributions of graph structured data, in [33] GPs were defined over graphs with undirected binary (positive or negative) edges and applied to semi-supervised learning problems. In [34] the authors applied GPs trained with variational approximations for semi-supervised learning on graphs that contain non-attributed edges. In [35] GP-based approaches are fused with deep learning for learning graph (e.g. network) structured signals.

# 4  Results

## 4.1  Wind farm operational data

A real-world industrial application, where relational structure is inherent in the observed data, is found in modeling of operational data of wind turbines positioned in a farm. The wind turbines (nodes) feature static variables, such as their power production characteristics and their position, as well as dynamic variables such as their current operational state. The actual operational state of a turbine is only known up to a certain precision from historical data, (i.e., Supervisory Control and Data Acquisition (SCADA) data), which is usually limited to 10 minute statistics. Due to the stochasticity of the wind excitation, compounded by incomplete information due to coarse measurements, there is *uncertainty* associated with the actual operational state of a wind turbine. Wind turbines arranged in a wind farm interact through the so-called *wakes*, which are travelling

vortices that affect the power production and vibrations of downstream turbines. The magnitude of wake effects is related to large scale turbulence (which is a global dynamic variable), to wind orientation (which is a global dynamic variable), to upwind turbine nacelle orientation (which is a node dynamic variable), the relative position between the two turbines (an edge static variable), the rotor diameter and the distance between the two turbines. The interaction is one-way directional but can change directionality depending on the wind orientation. The effect of wakes is stochastic due to turbulence. For robust wind power prediction, monitoring, control, and maintenance planning, we want to infer the distribution of operational characteristics of a wind farm conditioned on turbine characteristics and farm layout. Of crucial importance is the inclusion of stochastic variables in the interactions (i.e., edges) of the considered graph. Static graph edges, used as part of the graph conditioning, are constructed by considering the spatial proximity and relative position of pairs of turbines. The goal is to generalize directly to unseen farm configurations while learning directly on real condition monitoring data (zero-shot generalization) but at the same time to yield uncertainty estimates.

**Graph machine learning in wind farm modeling**   In [36] a GNN was trained on simulated data for wind power prediction. Recently, in [37] GNNs were applied as a surrogate model to more accurate fluid dynamic simulations. With the architectural advancements proposed in this work, we extend the wind farm relational modeling literature by providing a solution for representation of uncertainty in wind turbine interactions. Moreover, we empirically show in real wind farm data that significant accuracy improvements are possible through the incorporation of the proposed relational modeling and variational Bayes approach.

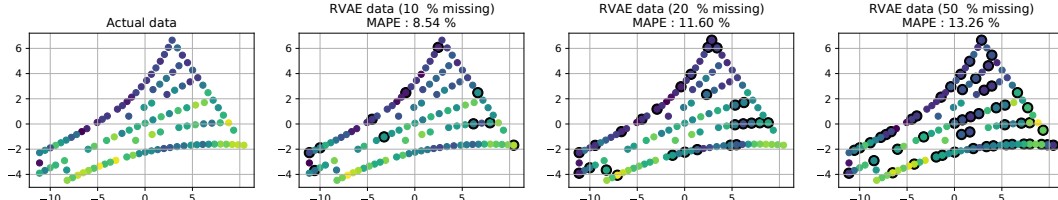

Figure 2: Imputation qualitative results for wind speed. The imputed points are marked with a dark circle on the background. The mean absolute percentage error is reported, which is computed as $1/N^T \sum_{i=1}^{N^T} (|\mathbf{v}_i^T - \hat{\mathbf{v}}_i^T|)/|\mathbf{v}_i^T|$ where $\mathbf{v}_i^T$ is the actual value of node $i$ , $N^T$ the number of target turbines and $\hat{\mathbf{v}}_i$ is the CRVAE prediction.

## 4.2   Real wind farm SCADA dataset

Conditional RVAE models (CRVAE) were trained with with a 80/20 train/test split on a dataset that includes 6 months of 10-minute average SCADA data readings. Since the goal is to compare the fitting capability of the models and not model selection, no validation set was employed. Early stopping with patience of 2500 steps was used (test set evaluation every 500 steps). The larger RVAE models that also yield the best performance had not converged at the 10th epoch. The 20% of turbine data are randomly masked during training. A batch size of 16 was used for all models. In order to make a fair comparison no regularization or KL-annealing was used. A small learning rate of $5 \cdot 10^{-5}$ and the Adam optimizer [38] with default parameters was used for all the runs. The final ELBOs for all models are shown in Table 2. A $mean$ aggregation function and composite aggregation function consisting of a concatenation of $mean$,$max$ and $min$ aggregators were used. Due to the concatenation operation, the composite aggregators result in slightly larger networks. Aligned with recent results on GN performance when using composite aggregation functions [39] we find that networks with the $mean - max - min$ aggregator indeed yield better performance. The motivation, however, for using composite aggregators, is also due to the physics of the problem. By using such aggregators it is easier to discriminate the un-waked part of the farm and the waked turbines. More concretely, turbines at the upstream boundary of the farm have larger power production and this directional effect can easily be masked using the mean aggregation. The CRVAE models are compared to a two-layer MLP-based CVAE trained with the arbitrary conditioning objective [30] of varying sizes, with the largest CVAE model number of parameters corresponding to the number of parameters of the best performing RCVAE. The largest CVAE model was the worst-performing of the evaluated CVAE models.

The CVAE model with the smallest size has slightly better performance compared to the RVAE model that performs no message passing on the encoder part, and therefore ignores relational inductive biases when inferring $G_z$. All but one of the CRVAE models strongly outperform the CVAE models by a large margin which is attributed to the effective use of relational inductive biases. To further support this claim, in the supplemental material (section A.1) gradient sensitivities are plotted and it is observed that the imputation results for masked turbines depend on upstream turbines. Qualitative imputation results are shown in Figure 2.

Table 2: Test set ELBO on Anholt SCADA dataset after 10 epochs. Numbers in parentheses are the standard deviations of the ELBO estimates in the test set (higher is better). The same node, edge and global latent sizes were used ($N^{G_z}$). "(comp.)" stands for the composite mean-max-min aggregator. All MLPs are 3 layer ReLU MLPs. The $\cdot^*$ superscript denotes results that were not derived from early stopping.

| Model | mlp units | $N^{G_z}$ size | MP Steps enc. | MP Steps dec. | agg. | # params | ELBO |
|-------|-----------|----------------|------|------|------|----------|------|
| CRVAE | 64 | 32 | 0 | 1 | mean | 184,717 | $1.96, (0.30)$ |
| | 64 | 32 | 1 | 1 | mean | 341,517 | $6.99(0.29)$ |
| | 64 | 32 | 2 | 2 | mean | 498,317 | $7.48(0.61)^*$ |
| | 64 | 32 | 2 | 2 | (comp.) | 522,893 | $\mathbf{8.11(0.48)}^*$ |
| | 64 | 32 | 3 | 3 | (comp.) | 679,693 | $7.70(0.53)^*$ |
| CVAE | 128 | 64 | – | – | – | 77,194 | $2.12(0.10)$ |
| | 256 | 64 | – | – | – | 252,554 | $1.17(0.16)$ |
| | 384 | 96 | – | – | – | 563,146 | $1.23(0.09)$ |

**Effect of inferring edge latents $\mathcal{E}_z$** The introduction of continuous edge-related latent variables is overlooked in a large part of the literature. Wake effect modeling is an application that may benefit from edge latent variables. We test the effect of edge latent variables by setting $\beta_{\mathcal{E}} = 0$ while still using $G_h$. The results of this experiment are shown in Table 3. The inclusion of the KL term with respect to edge latent variables seems to improve the reconstruction error achieved by the model.

Table 3: Effect of edge latent variables. Results based on 3 runs for each case.

| Case | $\log p(\hat{\mathcal{V}}_x | G_z; G_h)$ | Range |
|------|------|-------|
| $\beta_{\mathcal{E}} = 1.$ | $\mathbf{4.16}$ | $\pm 0.43$ |
| $\beta_{\mathcal{E}} = 0.$ | $1.80$ | $\pm 1.21$ |

## 4.3 Wind farm simulation dataset

The steady-state wind farm wake simulator FLORIS [40] was used. A dataset of wake effect simulations and preprocessing tools for demonstrating the wind farm modeling approach adopted herein is released as part of this work. In what follows we test the generalization capabilities of a trained RVAE to novel geometric configurations. A single farm configuration is used for training and another one is used for testing. Both farms are simulated with random wind characteristics such as direction and average wind speed. An example output from the simulation can be found in the supplemental material. The train and test farm configurations can also be found in the supplemental material.

**Qualitative results** The RVAE model is able to capture the orientation-dependent wake deficit for each turbine separately on the test wind farm as shown in Figure 3. Furthermore, we use a single turbine as a probe and position it on a regular grid while keeping a turbine on a fixed position (0,0). By inspecting the wind speed predicted at the probe turbine, we can map the wake deficit in 2D behind the source turbine. This is shown in Figure 4. The spatial dependence of the wake deficit is also shown as computed from FLORIS and the error in RVAE estimation. For distances larger than $200m$ the wind deficit is accurately predicted. Note that this result is from a model trained on operational data from a *single* simulated windfarm. When the turbines are very close ($< 200m$) the wakes are not predicted correctly, but this is an expected effect since the RVAE never encounters turbines at these distances. Wake effects estimated with the RVAE are slightly lower than those derived from the simulation as shown in Figure 3. However, the RVAE seems to capture the intricate wind orientation-dependent effects which depend on the farm layout.

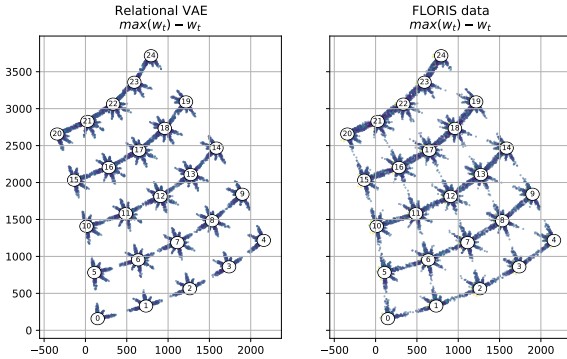

Figure 3: Wind deficits on the simulated test farm and estimates from the trained RVAE. Each point associated with a turbine is plotted in a 2D polar coordinate system centered on the turbine. Each point is plotted towards the orientation of the *incoming* wind. The distance from the origin is proportional to the wake deficit, estimated as $max(v) - v$ where $v$ is mean power and mean wind.

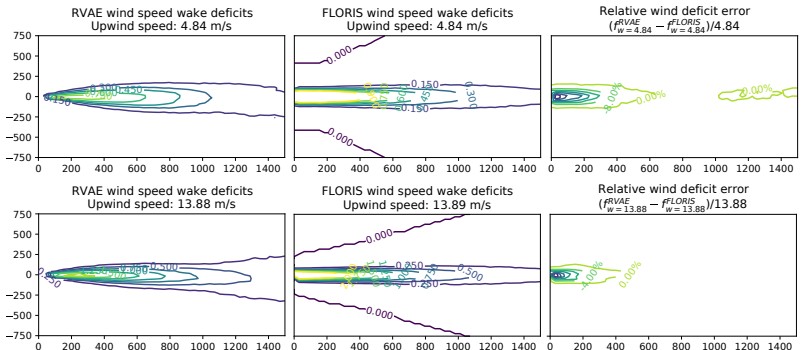

Figure 4: Learned spatial distribution of wake related wind speed deficit, evaluated as $w_{(0,0)} - w_{(x,y)}$ where $w_{(0,0)}$ is the the wind speed at the up-wind turbine and $(x,y)$ denotes the wind speed for a *probe* turbine positioned at $w_{(x,y)}$.

### 4.4  1D regression

In order to further demonstrate the versatility of the RVAE in modeling structured data, and in order to make the connection to NPs [18] clearer, in what follows an RVAE adapted for node data imputation is presented [30]. The dataset consists of sets of points sampled from a zero-mean 1D Gaussian process with a squared exponential kernel. The pairs of input points $\{(x_{1:n_m}, y_{1:n_m})_m\}$ are used as node features to construct a set of context and target graphs, where $m$ corresponds to different GP realizations. Each graph contains a set of edges $\mathbf{e}_i$ which encode the relative position of the observation points. The edge features between observations at points $x_i, x_j$ are defined as $f(x_i, x_j) = e^{-c \cdot |x_i - x_j|^2}$ where $c$ is a function of the cutoff distance for edge creation. Note that the construction of such edge features endows the model with translation equivariance. In contrast to MPNPs, [29], the edges are the same in the context and target graphs. The loss function used is the same as in Equation 10. Both $p_\phi$ and $q_\theta$ are implemented as GNs. The outputs of the GNs parameterize a Gaussian, i.e.,

$$q_\phi(z|\mathcal{D}) = \mathcal{N}(\mu_\phi(\mathcal{D}), \sigma_\phi^2(\mathcal{D})), \quad p_\theta(y|\mathcal{D}) = \mathcal{N}(\mu_\theta(\mathcal{D}), \sigma_\theta^2(\mathcal{D})). \tag{11}$$

The $y$ values of $\mathcal{D}^T$ are replaced with 0 when fed through the encoder and an additional binary feature $b$ for the node, which denotes masking, is appended to the node tuple. The $b$ feature is zero for the unmasked nodes and 1 for the masked nodes. The masked input is denoted by $\mathcal{D}^{T \setminus b}$. The union of the masked target input with the context dataset is denoted by $D^{T \setminus b \cup C}$. Instead of using two different functions for the prior of $p(z|\mathcal{D}^{T \setminus b \cup C})$ as in [30], and posterior network $q(z|\mathcal{D}^{T \cup C})$ and in order to

keep the conditional RVAE model closer to the NP formulation, the approximate posterior (i.e., the encoder of the RVAE) is used also for the learned prior. The decoder $p_\theta$ receives as node conditioning (and optionally edge conditioning) the $x_T$ values and the global latent variable $\mathbf{u}^z$. Each realization of $\mathbf{u}^z$ corresponds to a different context set which in turn corresponds to a different sampled $GP$. More information about he training setup can be found in the appendix.

The NP is implemented by defining a DeepSet encoder, a global latent variable of the same size as the NP MLP. The same latent variable size for nodes and edges was used for each experiment, which is the same as the core size. All aggregation functions are *mean* aggregations. Experiments were performed with different number of message passing steps, and inclusion of either the relative observation position as an edge feature or the absolute node position $x_i$ for each feature. The models are tested in un-seen GP realizations and the negative log-likelihood of predictions are reported in Table 4. The RVAE models compute edge, node and global variables. The test datasets contain points with $x \in [0, 1]$ and $x \in [1, 2]$ ranges in order to test the generalization capability of the proposed model in translation. Since the edge-blocks only ever receive translation equivariant inputs from the dataset, the RVAE models generalize well in the $x \in [1, 2]$ range. This is presented only as an example of how special equivariant inductive biases may be implemented in RVAE. It is observed that the full RVAE model does not perform well when only the node features are available. As with NPs, it was empirically found that models yield better results with more training.

Table 4: Test set log likelihoods on 1D GP regression with Conditional RVAE. The results are based on a set of 5000 unseen GP samples, each with 50 context and 50 target points. The models were trained only on points with x in the $[0, 1]$ range. Values in parentheses are standard deviations of the mini-batches. RVAE denotes a model where all latent variables are used (edge node and global).

| Model | size (mlp/$z$/MP Steps) | Only cond. on nodes $G_h = (\mathcal{V}_h, \cdot, \cdot)$ | | Cond. on edges and nodes $G_h = (\mathcal{V}_h, \mathcal{E}_h, \cdot)$ | |
|---|---|---|---|---|---|
| | | $x \in [0, 1]$ | $x \in [1, 2]$ | $x \in [0, 1]$ | $x \in [1, 2]$ |
| CRVAE | 64/64/0 | $-17.94(3.11)$ | $-24.59(4.10)$ | $0.33(0.04)$ | $-0.21(0.13)$ |
| | 64/64/1 | $-12.55(2.51)$ | $-9.79(2.51)$ | $0.36(0.07)$ | $0.08(0.06)$ |
| | 64/64/2 | $-$ | $-$ | $\mathbf{0.98(0.09)}$ | $\mathbf{0.67(0.08)}$ |
| NP | 64/64/NA | $-1.34(0.07)$ | $-11.13(3.08)$ | NA | NA |
| | 128/128/NA | $-1.08(0.11)$ | $-31.74(14.08)$ | NA | NA |

## Conclusions and broader impact

This work introduces an attributed graph approach to the probabilistic modeling of relations within entities and their properties. The approach is verified and validated on wake effect simulations and actual data from wind turbines placed within a wind farm; a characteristic example that may be modeled as a graph. We also find some connections to the NP literature which we demonstrate by adapting the proposed method to perform a typical NP benchmark which is 1D regression for GP data.

We introduce a method for data-driven wake effect modeling for wind farms that accounts for uncertainty. The proposed method fuses physical intuition, flexible function approximation through GNs, and variational Bayes through re-parametrized gradients. Better and more computationally efficient wake effect modeling can lead to improvements in terms of accuracy and computational efficiency in analysis for wind farm siting [41] farm layout optimization [42], wind farm control optimization [43] and ultimately power production improvements, as well as more robust to uncertainties maintenance planning. Ultimately, the aforementioned lead to wind energy being a more attractive clean energy solution.

Graph data are naturally used to model social, transportation and communication networks. Possible negative implications of any graph ML work relate to possible malicious uses of analysis in such networks, such as de-anonymization in social networks [44], and vulnerability exploitation on transportation networks.

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
