# OpenReview forum: "Relational VAE: A Continuous Latent Variable Model for Graph Structured Data"
_NeurIPS.cc/2021/Conference — NeurIPS 2021 Submitted_

### Official Review · Reviewer_UzkQ · 2021-07-14

**Rating:** 4
**Confidence:** 2

**Summary:**

This paper proposes a framework for a general probabilistic generative model for attributed graphs.
The proposed model assumes that the nodes, edges, or global attributes of a graph have many different properties, which can be deterministic or stochastic, and observable or unobservable, and so on.
The proposed model formulates a stochastic dependency between the conditioning graph variable G_h and the state graph variable G_s similar to beta-VAE, assuming the existence of several latent variables in addition.
The paper shows that the proposed model encompasses many existing Bayesian graph models and evaluate its performance using wind farm dataset and artificial simulation dataset with ELBO or test data log-likelihood.

**Ethical Concerns:**

No specific concerns.

**Limitations And Societal Impact:**

No specific comments

**Main Review:**

## Originality:
In Sec. 2, the proposed model is explained with many equations. As far as I understood, the variables of the graph data are separated into conditioning and state, and the probabilistic dependency between these two groups of variables and latent variables is assumed to have a dependency structure similar to that of beta-VAE.
As far as I know, no VAE-like graph model has ever been proposed to model the process of graph data generation in such a general way, so there is a  concrete novelty at this point.

However,  it is unclear that what kind of benefits we can expect from the proposed model. This is because there are no observations or assumptions about the data to be modeled. In turn, there are no new ideas or inspirations to explain the characteristics of the data efficiently in the graphical models.
I felt that there is no strong answer given to the question, "We have created a new model, but is it beneficial?" throughout the paper.

## Quality

I think there is a room for improving this manuscript as I write in the following  "clarity" and "significance" issues.

 ## Clarity

In the introduction, the motivation for this research is not explained intensively. Therefore, it is difficult to answer "Why is this research necessary? What is the value of this research?''
Therefore, readers may find it difficult to understand these importantquestions. This problem is not resolved for me even after reading the paper to the end.

In L17-18, it is written ''the ubiquity of noisy structured data and systems with stochastic or partially observable
 interactions of industrial relevance (e.g. wind farms and urban transportation networks)''.
For most readers, this sentence alone does not explain the motivation of the study at all. This is because the readers are unfamiliar with these examples and have no knowledge of what they want to do with these data.

(A) what kind of uncertantiies exist in these data
(B) what kind of problems caused by the uncertainties
and (C) why existing GNNs and graph generation models are insufficient to solve them
Given concrete explanations of all A, B, and C, then, I think we can finally reach the understanding that a general form of a probabilistic graph generation model is necessary.

The same difficulty in understanding due to lack of explanation affects each technical discussion in Sec 2.
- L37-65 Why did you make the settings for each attributes so general (stochastic/deterministic, observable/non-observable, static/dynamic)?
In many DNN/GNN researches, it is of interest to choose appropriate inductive biases into the model to perform good at chosen tasks or goals. The formulation of Sec. 2 seems to go in the opposite direction, and I don't understand the intention or the motivations for a general formulation like this.

In Sec. 4, the examination of specific data begins.  Unfortunately, the explanation is not concise and easy to understand.
Since the paper introduces many symbols and conditional probability distributions in Sec. 2, I would like to describe the data with these symbols or equations:  for example, which symbol corresponds to an element of wind fard data, and which term of the conditional probability distribution will be important.
Clearer description of the data wouldl help to convey the appropriateness of choosing counterparts (CVAE and NP) for experiments. In the current manuscript, it is very difficult to understand the appropriateness of the choices.

## Significance:
The lack of explanation of motivation described in the Quality section makes it difficult for me to evaluate the significance of this paper. If we do not fully understand the reason ''Why is this study necessary?'', it is difficult to evaluate the significance of the content of the paper.

I think that the current experiment is insufficient to experimentally demonstrate the usefulness of the proposed RVAE.
(I) The proposed RVAE model is quite broad and general in its formulation.
(ii) The effective latent variables and conditioning variables are expected to be different from dataset to dataset.
Given (i) and (ii), it is necessary to conduct a thorough ablation study on many different datasets to evaluate the proposed method accurately.
By testing such experiments on several real-world datasets with different characteristics, we can see that RVAE is highly effective on any dataset, thanks to the general formulation.

Thus, in my opinion, the paper needs more experiments in various datasets. Ideally, the counterpart models should variy across dataset to dataset reflecting the different nature of datasets and tasks.

Figs. 3, 4 I think that it is insufficient to convince the readers of the usefulness of the proposed method even qualitatively unless the visualization of the estimation results by other methods is included in addition to the estimation results by RVAE.

(+) a broad and general formulation for VAE-like graph generation models
(--) motivations are not explained well
(-) description of datasets can be improved clearer and more informative way
(--) experimental validations are insufficient to show the superiority of a proposed general framework

## After author feedback

Thank you authors for the detailed clarifications.
Your feedbacks resolves several of my misunderstandings. Thus I raise my scores +1.

At the same time, the feedbacks reveal the manuscript lacks these important explanations.
Also I still believe the experimental validations are not sufficient for convincing non-expert readers like me.
Therefore I cannot vote for acceptance before confirming the revised manuscript fulfills these requirements.



**Time Spent Reviewing:**

6

---

> ### Author Response · Authors · 2021-08-10
> **Response to reviewer**
>
> > # Summary:
> "This paper proposes a framework ... and so on. ..."
>
> We wish to add a clarification to the reviewer's summary:
> * The model is not *a generative model for graphs* but, more precisely, it is a model for learning a probability distribution over attributed graph-structured data.
>
> >" ... The proposed model ... similar to beta-VAE,"
>
> In our work the $\beta$ factors are mainly for optimization purposes. There is no direct connection to $\beta-$VAEs, yet we claim extensions are easily possible.
>
> >"... assuming ... ELBO or test data log-likelihood."
>
> Indeed, using the proposed graph-structured latent and a graph-structured conditioning allows for flexible modeling. We would also like to add that we draw a connection to NPs (Neural Processes).
>
> ># Main Review:
> >## Originality:
> "...As far as I understood,...conditioning and state,..."
>
> This indeed holds for the input to the model. The conditioning $G_h$ is also passed to the decoder. We show that by doing so, we can retrieve a NP-like model.
>
> >"However, it is unclear ..  about the data to be modeled."
> > "...new ideas or inspirations ... data efficiently in the graphical models."
>
> The main benefit of the proposed framework is that it brings together graph deep learning and variational Bayes (VB) in order to allow for flexible fusion of known structure and known influencing factors.
>
> > "is it beneficial?"
>
> One clear claim we make in lines 224-232 is that the inclusion of the KL term with respect to edge latent variables facilitates a better performance in terms of the achieved log-likelihoods. This is also supported by the good performance of the Conditional RVAE (CRVAE) in Table 2.
>
> We finally further claim that the paper offers a further significant contribution in clarifying connections between different NP/GraphVAE models.
>
> > "In L17-18, it is written ''the ubiquity ... " no knowledge of what they want to do with these data."
>
> We can clarify this part in the final version. The proposed approach offers new routes for general applications of ML where it is important to take into account graph structure (relational data).
>
> For the farm data example, we present how the problem fits in this category in lines 174-185 in the manuscript. We decided to delay the discussion on explicit applications in order to introduce the methodological contribution of the proposed approach. For the same reason we introduce the notation before the "Related work" section.
>
> > " (A) what kind of uncertanties ... probabilistic graph generation model is necessary."
>
> * (A) this point is of course, problem-specific. Deep latent variable modeling (such as VAEs) relieves the burden of carefully specifying uncertainties. This is the motivation for using a deep Bayesian modeling approach in this work.
>
> * (B) The goal is to show that a conditional distribution over continuous data, that are defined over a graph, $p(G_x; G_h)$, can be learned and that this result can generalize to unseen configurations (figure 3). For the wind farm example, which is largely motivated from the problem of condition monitoring, the learning of such a baseline and generalizable distribution is useful for subsequently predicting deviations from this expected distribution.
>
> * (C) When using a NP encoder (no message passing), we show for the farm example in table 2, that the model performs like a simple VAE. When introducing message passing, performance increases. This is a strong indication that it is important to use the relational structure of the problem. We are offering a general view that allows the combination of GNs with VB (table 1). Architecturally, we make a case for including edge conditioning and edge latent variables.
>
> > " The same difficulty in understanding ... Sec 2. "
>
> Indeed, this section is condensed and not self-contained. On the other hand, reviewer **zzT8** found that this section was too basic. For readers not seasoned in graph ML, it serves as an introduction to the employed notation. However, we share this reviewer's point that entering this section without previously motivating the relevance for our chosen use cases is problematic and we intend to revise this part, by offering a smoother transition in Section 1.
>
> > " L37-65 Why did you make the settings for each attributes so general ... appropriate inductive biases into the model to perform good at chosen tasks or goals."
>
> We comment just below why these notions are introduced abstractly at first.
>
> > " The formulation of Sec. 2 seems to go in the opposite direction, and I don't understand the intention or the motivations for a general formulation like this."
>
> We decided to introduce the notation first and then the related work section because it is not possible to argue about "node/edge/global" conditioning and latent variables before introducing them. Also, without introducing the notation first, it is difficult to make the argument that the NP model is related to our model. This could have been better exposed with more discussion, but to satisfy the space constraints of the conference and make a relatively self-contained manuscript, it was decided to make this condensed exposition at the expense of making the manuscript less accessible to readers that are not familiar with GNs, NPs, and deep VB literature.
>
> > " In Sec. 4,... wind farm data..."
>
> Thank you for the opportunity to clarify this matter; we understand that the reviewers are not required to inspect the supplemental material, but we do release two jupyter notebooks that could have served in clarifying exactly this question. We detail this in what follows:
>
> ``` python
> # When a channel is completely missing, a NaN is entered in its returned values.
> UNIVERSAL_COL_NAMES= ['mean_wsp','mean_power', 'cos_phi', 'cos_yaw','sin_yaw', 'rated_power','rotor_diam','ambient_temp','wsp_std','is_masked']
> ```
> where
>
> `['rated_power','rotor_diam']` $\in \mathcal{V}_h$ and
>
> `['mean_wsp','mean_power','cos_phi','cos_yaw','sin_yaw','ambient_temp','wsp_std']` $\in \mathcal{V}_x$.
>
> $b$ = `'is_masked'`.
>
> The edge features are assigned in line 90 of the same file as
>
> ```python
> ...
> feats[t1,:,:] = np.array([sp,cp,d]).T
> ...
> ```
>
> where `['sp','cp','d']` $\in \mathcal{E}_h$
> are $[ \\sin(\\phi),\cos(\phi), d]$
>
> as also shown in figure 3 of the accompanying document in the supplemental material.
>
> ># Significance:
> "The lack of explanation ... significance of the content of the paper."
>
> We hope that with our comments above, we managed to better explain and motivate our work.
> Drawing connections from VB on sets (NPs), conditional VB with arbitrary conditioning, and GNNs, the RVAE is exposed with the hope that future researchers will be inspired for original and creative new applications of deep learning. We believe there are several interesting extensions and their properties that become clear when considering the exposed framework.
>
> The industrial application explored (wind farm modeling) with both real and simulated data, is both interesting and has potential for positive social impact. We believe that the great improvements in fitting performance (Table 2) and the (scalable) treatment of uncertainty in the derived models using amortized variational inference with a conditional VAE-like model is of practical value.
>
> > "I think that the current experiment is insufficient to experimentally demonstrate the usefulness of the proposed RVAE."
>
> > "(I) The proposed RVAE model is quite broad... datasets and tasks"
>
> Since the model unifies several well-motivated approaches (NPs/Variational Bayes on Graphs), we find that the numerous ablation studies, which have already been conducted within the context of the works that we cite, already explain how the properties of each model change when different parts are omitted (see also Table 1). The ablations performed were highly targeted.
>
> Namely,
> * using a GN-inference model against using an MLP (Table 2), confirming the usefulness of the proposed application of VAE-based GNs (conditional RVAE) on farm data,
> * enabling/disabling the KL regularization term for the edges (Table 3), confirming the usefulness of edge-bound KL regularization, and
> * allowing the network to learn translation equivariance by providing translation equivariant features and surpassing the performance of NPs (we do recognize that there are techniques in the literature, such as AttCNP and ConvCNP which may be better for that task, yet our approach is more flexible)
>
> > Figs. 3, 4 insufficient to convince the readers...
>
> The setting presented in Figs. 3 and 4. require a model that can take into account directional effects between turbines conditioned on wind orientation for an arbitrary number of turbines and jointly taking into account the effect of wind orientation and relative turbine position. We believe it is hard to find a competitive alternative to the proposed model.
>
> > (+).. broad and general formulation...VAE-like
>
> We are glad that the reviewer recognizes this point.
>
> > (--) motivations are not explained well
>
> We hope that we partially addressed this in our comments above.
>
> > (-) description of datasets can be improved clearer and more informative way
>
> We are willing to improve upon this front. However, we open-source dataset and source code.
>
> > (--) experimental validations are insufficient to show the superiority of a proposed general framework
>
> We make no general claim on the superiority of our method. We do claim that by specializing the model (table 1) one retrieves important properties covered by other models (NP/GraphVAE/NRI).
>
> ## Closing
> We want to thank the reviewer again for the detailed review. We regret that the reviewer did not find the manuscript of sufficient quality, importance, and originality for publication. In reciprocating the effort that the reviewer put, we addressed major points in detail in our comments above. One point we cannot readily address is a large-scale ablation (computational restrictions), but we also think it is not entirely necessary.

---

> > ### Comment · Reviewer_UzkQ · 2021-08-15
> > **Thank you for your comments**
> >
> > Authors,
> > thank you for your detailed comments. It seems I need to re-examine the manuscript with the help of your comments.

---

### Official Review · Reviewer_zzT8 · 2021-07-15

**Rating:** 5
**Confidence:** 3

**Summary:**

This paper proposes a generalized formulation of a probabilistic model for graph-structured data that incorporates graph networks (GN) (or rather, provides a probabilistic extension thereof). The authors show that the proposed framework is general enough to subsume existing approaches such as Neural Relational Inference (NRI) and Neural Processes (NP). The method is empirically validated on simulated and real-world data, with emphasis placed on the modelling of wind farm data.

**Limitations And Societal Impact:**

The authors recognize the potential negative societal impacts of their work, in particular, its potential malicious uses in de-anonymization of social networks and exploiting vulnerabilities in transportation networks.

**Main Review:**

This paper is generally well-written and reasonably easy to follow. There are a few grammatical mistakes that commonly occur throughout the paper; a non-exhaustive list is provided in the Misc Issues section of this review. I checked the details of the mathematical equations and was convinced that the proposed framework is technically sound.

However, it was difficult to assess the significance and originality of this paper's main contribution, and I was not totally convinced. The proposed framework is indeed quite general and shown to include existing methods such Neural Relational Inference (NRI) and Neural Processes (NP) as special cases. On one hand, generalizations such as this are often important contributions since by providing a framework under which to unify existing approaches, they can help provide a better understanding of the existing approaches, in particular how and why they work, and clarify the conditions and assumptions under which they are closely related. On the other hand, there is a fine line between generality and vagueness, the latter case being that something is so general it ceases to be particularly insightful or interesting.

In Section 2 "Methods", there is nothing particularly novel and original. The framework formalizes the various components of a graph -- the nodes, edges, and global attributes, and considers their various taxonomies as deterministic or stochastic, observed or hidden, dynamic or static. A subset of the observed components makes up the conditioning or context variables and another makes up the state variables. A graph-structured latent variable consisting of vertices, edges, and global variables is assumed. A joint distribution over these variables is then specified, which factorizes into distributions over the vertices, edges, and global variables, respectively. All of this is straightforward and fundamental to probabilistic modeling.

> Sidebar question: Eq (3) what specifically are these prior distributions? I couldn't find these specified anywhere.

Next, the procedure for approximate inference over the graph-structured latent variable is described. Again, this is just a description of the standard application of the principles of variational inference using amortized inference with a Graph Network, with the addition of scaling the KL divergence terms using a $\beta$ hyperparameter as in $\beta$-VAEs.

> Sidebar comment: line 76-77 "A GN is proposed for inferring the parameters." - in more precise terms, you're *amortizing* the inference on these parameters using GNs. Consider perhaps invoking this term to make this more explicit.

Regarding the experimental results, there is a concern that the method might be overfitted to wind farm data, and might fail to generalize well beyond this context. On the one hand, I was pleased to see the method applied to a real-world problem, particularly one that has a high potential for positive societal impact (contributing to better clean energy solutions). On the other hand, as a methodological contribution, I would have liked to see results on a more diverse range of problems, both real and synthetic. The other issue is the lack of comparisons against baselines. On the real wind farm dataset, the CVAE is the only baseline considered. On the 1D regression dataset, NP is the only baseline considered.

In summary, I find the contribution to be fairly significant but somewhat incremental. In particular, the originality and novelty are somewhat limited. The paper is communicated clearly and of reasonably high quality. The experiments are mostly well-executed but may be over-specialized to the domain of wind farm modeling.

### Misc Issues

- line 212: "The CRVAE models are compared to a two-layer MLP-based CVAE trained with the arbitrary conditioning objective [30]" - this is the first occurrence of the acronym "CVAE". From the context and subsequent citation, it's possible to guess that it refers to the conditional variational autoencoder, but the reader shouldn't need to guess.
- Please consider rasterizing the figures on Page 8 (the culprit is most likely Figure 3) which causes my PDF viewer to lag on a reasonably powerful computer.
- line 309, reference: please sort your references alphabetically by the first author's last name
- line 328, citation [7]: "bariational bayes"

The following sentences all have extraneous commas (probably non-exhaustive):

- line 60: "The full graph state, is denoted by"
- line 114: "in this work, the above mentioned approaches, are generalized and unified"
- line 128: "the input-output observation cases D, are split as"
- line 146: "The aforementioned models, feature a global latent variable"
- line 209: "for using composite aggregators, is also due to the physics of the problem"



**Time Spent Reviewing:**

6 hours

---

> ### Author Response · Authors · 2021-08-10
> **Response to reviewer comments**
>
> In what follows, we address in detail the points raised by the reviewer. We would like to express our gratitude the thoughtful and high-quality review.
>
> > ## Summary:
> > This paper proposes ... wind farm data.
>
> The summary of the reviewer is accurate.
>
> > ## Main Review:
> > This paper is generally well-written and reasonably easy to follow.... convinced that the proposed framework is technically sound.
>
> We will correct the grammatical errors; thank you for pointing that out.
>
> > However, it was difficult to assess the significance and originality ... not totally convinced.
>
> We acknowledge that the contribution was not stressed enough in the introductory section, which we intend to update, for clarifying that the contributions of this work are:
> * formalizing the commonalities between the NP, Bayesian GNN and CVAE literature,
> * demonstrating a general CVAE/NP using Graph Networks
> * demonstrating the application of the model in 2 synthetic and 1 real-world dataset of industrial/environmental relevance
> * showing highly targeted comparisons to baselines for the proposed model (CVAEs/NPs), demonstrating the effectiveness of the model
> * showing generalization qualitatively (fig. 3&4) and quantitatively generalization (tables 2 and 4) properties.
>
> >The proposed framework is indeed quite general and shown to include existing methods such as ... can help provide a better understanding of the existing approaches...
>
> Indeed we hope that the offered view on the treatment of uncertainty in GNNs and making a connection to NPs and CVAEs contributes towards making connections between the different fields touched upon and that it inspires future extensions. We believe that the paper does deliver in that respect for the literature covered and for the features of Bayesian GNNs/NPs covered.
>
> > On the other hand, there is a fine line between generality and vagueness...
>
> We made a conscious effort to limit the literature review to models that do not deviate from the paradigm we introduce to avoid stretching the arguments beyond their validity and introduce vagueness.
>
> > "In Section 2 "Methods", there is nothing particularly novel and original. ... probabilistic modeling."
>
> We included this part (lines 32 to 51) for completeness and to introduce notation. From line 52 onwards, the general model is introduced. We will split the section to "**preliminaries**" and from 52 onwards as "**methods**".
>
> >Sidebar question: Eq (3) what specifically ...
>
> Here, there is indeed a point that needs clarification. The distribution over a graph is assumed as a factorized distribution over the nodes, edges, and global attributes of the graph. More specifically, for a graph $G = (\mathcal{V}, \mathcal{E}, \mathbf{u})$,
>
> $$
> p(G) = \\prod_{i=1}^{N^v} p^{\\mathcal{V}} ({\\mathbf{v}_i}) \prod^{N^e}_i p^{\mathcal{E}} (\mathbf{e}_i) \cdot p^{\mathbf{u}} (\mathbf{u} )
> $$
>
> For a RVAE with no conditioning (i.e. no $G_h$) one would take, for instance, a Gaussian prior over each latent and retrieve the KL objective of the VAE for each component (edges, nodes, globals) separately. When graph-structured conditioning $G_h$ exists, which can be some static properties the node, edge, and global (for the farm example, turbine power rating for the node, relative geometric position for the edges) that conditioning needs to be taken into account when decoding the variables. One may use $G_h$ to the prior or to the decoder as shown in the RVAE in figure 1. When adding conditioning to the decoder, one may more accurately write,
>
> $$
> p(G_z;G_h)  = \prod_{i=1}^{N^v} p^{\mathcal{V}}(\mathbf{v}^z_i;\mathbf{v}^h_i) \prod_{i=1}^{N^e}p^{\mathcal{E}}(\mathbf{e}^z_i;\mathbf{e}^h_i) \cdot p^{\mathbf{u}}(\mathbf{u}^z;\mathbf{u}^h) \\\\
>  = \prod_{i=1}^{N^v} p^{\mathcal{V}}(\mathbf{v}^z_i) \delta(\mathbf{v}^h_i) \prod_{i=1}^{N^e}p^{\mathcal{E}} (\mathbf{e}^z_i)\delta(\mathbf{e}^h_i) \cdot p^{\mathbf{u}}(\mathbf{u}^z)\delta(\mathbf{u}^h)
> $$
> with $\delta(a)$ denoting a Dirac-delta distribution on $a$, and a spherical unit Gaussian for the rest of the variables
> $$
>     p^{\\mathcal{V}}(\\mathbf{v}^z_i) = \mathcal{N}(0, \mathbf{I}) \in \mathbb{R}^{d_v},\quad
>     p^{\\mathcal{E}}(\\mathbf{e}^z_i) = \mathcal{N}(0, \mathbf{I}) \in \mathbb{R}^{d_e},\quad
>     p^{\\mathbf{u}}(\\mathbf{u}^z) = \\mathcal{N}(0, \mathbf{I}) \in \mathbb{R}^{u}
> $$
>
> When using the *arbitrary conditioning* VAE ([Ivanov 2018](https://arxiv.org/abs/1806.02382)), where the prior is also learned, these distributions are Gaussian MLPs, that use the node/edge/globals from the encoder as in equations 5,6,7. Thank you for pointing that out and we will include this clarification in the final version.
>
> >Next, the procedure for approximate inference over the graph-structured latent variable is described. Again, this is just a description ...
>
> Indeed this is the case. This part may be valuable for readers that are less comfortable with the concepts introduced.
>
> >Sidebar comment: line 76-77 "A GN is proposed for inferring the parameters." - in more precise terms, you're amortizing the inference ...
>
> That is an excellent suggestion which we will adopt.
>
> >Regarding the experimental results ... (contributing to better clean energy solutions).
>
> As the reviewer observes, the method is related to NPs and yields comparable/better results (when including edge features). Consider the implementation of invariances/equivariances in NPs. For the 1D GP regression example, we manage to easily retrieve translation equivariance by using translation equivariant edge features. This would not have been possible with a NP since we cannot encode the relations in $x$ between the context points. That is the main motivation behind ConvCNPs which, however, is treated in our work in a much simpler manner by making at the same time the connection to GNs.
>
> Considering other real-world applications of the proposed model, one can envision latent variable models for traffic data or epidemic spread which also feature a graph structure and have complex hidden dynamics. We believe that the good alignment of GNNs to inference on graph-structured data and the fusion of observable conditioning and deep models will yield exciting applications in such problems.
>
> > On the other hand, as a methodological contribution, ... results on a more diverse range of problems, both real and synthetic.
>
> We will keep this suggestion in mind for potential revisions or future work. However, we would like to point out that the subsumed model were already proven useful in a variety of real and synthetic examples.
>
> >The other issue is the lack of comparisons against baselines. On the real wind farm dataset, the CVAE is the only baseline considered. On the 1D regression dataset, NP is the only baseline considered.
>
> An important property of the model is that through the graph conditioning (in particular edge conditioning $\mathcal{E}_h$), it can generalize to unseen farm geometries (figure 3). A toy synthetic problem adaptation demonstrating this is 1D GP regression. Due to computational budget constraints and in service of conciseness, we had to make a strict selection of baseline models motivated for benchmarking both problems. A large-scale analysis of what parts of the model are necessary for different problems is indeed well-motivated, but we hope that this model enables the flexible fusion of learned and imposed structure for variational Bayes (NPs/CVAEs), the same way the [Graph Nets framework](https://arxiv.org/abs/1806.01261) enables that fusion in predictive models.
>
> >In summary, I find the contribution to be fairly significant but somewhat incremental. In particular, the originality and novelty are somewhat limited. The paper is communicated clearly and of reasonably high quality. The experiments are mostly well-executed but may be over-specialized to the domain of wind farm modeling.
>
> We thank the reviewer for finding the contribution significant. We hope that the perceived incrementality of the contribution is balanced by the fact that our work makes several original connections to seemingly unconnected parts of the ML literature. We hope that the view offered in this work will contribute to creative fusion between techniques and inspire applications of GNNs and Bayesian deep learning. Since two-thirds of the experiments are in wind farms, we can see how it may come out that the method is specialized to wind farms. We claim, however, that having unobserved edges with both latent and state variables is a very common problem that is rarely treated. One can easily devise use-cases where inference over edge-bound latents (which is the main claimed advancement with respect to the prior literature, besides the unified view of NPs and variational Bayesian approaches that use GNNs) can be useful in social network analysis, disease spread in networks, traffic networks, and computer networks.
>
> > ## Misc Issues
> > line 212: "The CRVAE models are compared to a two-layer MLP-based CVAE trained with the arbitrary conditioning objective [30]" - this is the first occurrence of the acronym "CVAE". From the context and subsequent citation, it's possible to guess that it refers to the conditional variational autoencoder, but the reader shouldn't need to guess.
>
> Indeed, this is unclear and will be corrected.
>
> >Please consider rasterizing the figures on Page 8 (the culprit is most likely Figure 3) which causes my PDF viewer to lag on a reasonably powerful computer.
> line 309, reference: please sort your references alphabetically by the first author's last name
> line 328, citation [7]: "bariational bayes"
> The following sentences all have extraneous commas (probably non-exhaustive):

---

> > ### Comment · Reviewer_zzT8 · 2021-09-04
> > **Acknowledgement**
> >
> > I want to thank the authors for providing a thorough response, and for helping to address some of my questions and concerns about their submission. In my review, I raised the concern that the description of the method while general, might be somewhat vague, and therefore difficult to understand specific important details about what is being proposed. From discussions with the other reviewers, particularly with Reviewer 2qz2, I was made aware that there are some crucial aspects that have not been given adequate treatment. For example, one critical point is how the proposed model guarantees exchangeability and consistency. Given the number of outstanding issues that still exist, I doubt it is possible for a minor revision of the manuscript in its current state to address these adequately. Therefore, I am disinclined to revise my overall rating.

---

### Official Review · Reviewer_2qz2 · 2021-07-16

**Rating:** 4
**Confidence:** 5

**Summary:**

This paper introduces Relational VAE, which is an extension of NPs for graph-structured data. However, the selling point of the paper is not that much clear.

**Limitations And Societal Impact:**

The complexity of the model and dealing with large graphs can also be considered as the limitation of the model.

**Main Review:**

The paper is hard to follow, motivation is not very clear, and seems to have some notational mistakes. I had to read it multiple times and still do not know what exactly the authors plan to do with their model.

I also believe the authors change some ML notations which are not necessary and sometimes are incorrect, starting from the title. For example, they mentioned their model as Relation VAE, however, they really proposed a new version of NP for graph-structured input data. Even from the lens of VAE, the relation in this model is the input graphs. As stated in lines 117-120, the authors stated " Inferring graph connectivity or generating graphs, however, falls out of the scope of this work. In RVAE the focus is generative modeling of graph structured data with an apriori known connectivity, with [...]." Or they clearly could use index $c$ instead of $h$ to refer to the context/reference points.  The article has the unfortunate tendency to contains a few unproven (or wrong) claims. For instance, "In this work, the above mentioned approaches, are generalized and unified in the proposed Relational Variational Autoencoder (RVAE) model", which would deserve some empirical/theoretical evidence. For example, the goal of NRI, is inferring the relation between objects in dynamic systems, where the input might not be structured. I cannot see how the proposed model is extending this.

My main concern with this paper is novelty: the method seems to be a straightforward application of NP/CVAE to a different data domain on which variational auto-encoders are being used. Comparing to NP, this paper only considered the observed samples are graphs. So, they only need to change the architectures to GNNs. For example, Equation (4) is similar to NP equation in [ Garnelo et al, NeurIPS 2018].

Concerning the experiments, I am not an expert in wind farm monitoring data, but I believe that as an ML conference, the authors do not clearly define and motivate the problem for a reader. Besides that, they could compare with some other existing baselines or include some sort of ablation study.

Overall, I think that this paper would need to be extremely solid on the experimental side with potentially further experiments, e.g. on node classification and more detailed analysis with respect to label rates of nodes, robustness to missing edges etc, in order to make up for the somewhat limited contributions in terms of novelty. Apart from that,  plotting $G_z$, showing the benefit of uncertainty modeling in the proposed method using pavpu or etc. would be helpful. For now, I cannot recommend this paper for acceptance at NeurIPS.


There are multiple notational mistakes.

1) = in equation 10 is missed in the second line
2) I think the ; in $p_\theta(G_z; G_h^{(i)})$ in equation 4 is not correct
3) I believe an expectation is missed in Eq 4
4) writing a loss for each sample in Eq 4 is not meaningful, while $G_z$ does not have any index.

If the model is Bayesian Graph modeling, the authors need to discuss and compare it with other Bayesian Graph modelings as well.
[1] Bayesian Graph Neural Networks with Adaptive Connection Sampling, ICML 2020
[2] Bayesian Graph Convolutional Neural Networks for Semi-Supervised Classification, AAAI 2019
[...] There are multiple other related papers

Equation 6 assumes the normal distribution for the relations. This might lead to a very dense graph.

I cannot follow lines 91 - 95 in that place. You might want to put them in the discussion or related works.

Validation could be used for early stopping. It is not clear to me how you used test data for this.

It is not clear how the authors change the name of the model from RVAE to CRVAE and what are the differences.



**Time Spent Reviewing:**

4 hours

---

> ### Author Response · Authors · 2021-08-10
> **Response to review comments**
>
> We would like to thank the reviewer for the detailed and thoughtful review. We find that it raised some interesting points but also contains some misconceptions we address in what follows.
>
> ## Summary:
> > *"This paper introduces Relational VAE, which is an extension of NPs for graph-structured data...."*
>
> Although technically, this is not a wrong statement (that some NPs and extensions can be cast in some sense as a Conditional RVAE) this is not the only thing we try to convey in this paper. The paper offers connections between Conditional VAEs, NPs and variational Bayes (VB) on graphs.
>
> >"However, the selling point of the paper is not that much clear."
>
> The “selling point” of the paper on the methodological front is offering the connections mentioned in the previous remark.
>
> On the practical front, motivated by a specific problem (i.e., the modeling of wind farms), an extension of latent variable models for graphs that introduce continuous latent variables for edges is proposed. An application is presented where this is of significant utility (as evidenced by Tables 3 and 4) and, in addition, it is shown that the model can make effective use of translation equivariant edge features to learn the translation equivariance of the 1D GP regression problems, which is the strong point of ConvCNPs (Table 5, implementation demonstrated in the notebook code provided).
>
> ## Main Review:
> > ... hard to follow, motivation is not clear, ...notational mistakes...
>
> The motivation of the paper is general supervised/semi-supervised learning for continuous graph-structured data. We observe that recent approaches to Neural Processes and to Variational Bayes for graphs have overlapping and complementary modeling strengths, summarized in table 1 (not an exhaustive list). We do believe this connection is valuable since it connects seemingly unconnected approaches. The proposed model, which introduces edge-bound latent variables (also introduced in a discrete setting in NRI), finds application in learning latent variables over wind farm states. This was indeed the initial inspiration for extending graph ML to contain edge-latent variables and this model. Allowing the inference model to infer latent variables over edges (as opposed to having only global and/or node-bound latents as the majority of the literature as summarized in table 1) is motivated by wake effects.
>
> What the proposed model allows for are the following:
> * Better performance due to exploiting the graph structure that underlies the problem (table 2)
> * “Local” (i.e., node) conditioning (as AttCNPs  and DSVNP).
> * Implementing invariances, such as translation invariance, through the implementation of edge feature invariance, as ConvCNPs as evidenced by the good performance in NLL for the 1D regression example when training with $x \in [0,1]$ and testing with $x \in [1,2]$ (also shown with ConvCNPs)
>
>
> > I also believe the authors change some ML notations which are not necessary and sometimes are incorrect, starting from the title. For example, they mentioned their model as Relation VAE, however, they really proposed a new version of NP for graph-structured input data.
>
> We would like to argue that both terms are correct. Unfortunately, the names "Graph Neural Processes" and Message Passing Neural Processes were already used for other models.
>
> > Even from the lens of VAE, ... context/reference points.
>
> Here we believe there is some misunderstanding from the side of the reviewer on how the NPs are implemented in our framework. In the manner the proposed model operates, these are *not the same*. For instance, for the 1D regression problem, the points in the $x_T$-axis are used as a node conditioning on the decoder side. Therefore, $x_T \in \mathcal{V}_h$. See also line 273, where an explicit mention is made to this.
>
> > The article has the unfortunate tendency to contain a few unproven (or wrong) claims. ... empirical/theoretical evidence.
>
> Please take note of the computational diagram of NRI and NP are offered in Figure 1, together with correspondences between these models and the components of the more general RVAE. We hope this serves as a good starting point for realizing the connections of the models cited in Table 1.
>
> >For example, the goal of NRI, ... I cannot see how the proposed model is extending this.
>
> Equation 4 in our paper is a generic form of the ELBO when constructing models for graphs. Specializations of this equation, with only nodes and globals retrieves the ELBO that gives NPs (see table 1). This can be found in several works in the literature. Regarding (Garnello 2018) we assume the reviewer refers to [this work](http://bayesiandeeplearning.org/2018/papers/92.pdf) which does not touch upon the claimed architectural or methodological novelty of this work. (edge latents, general variational Bayes for attr. graphs).
>
> > Concerning the experiments, ... some other existing baselines or include some sort of ablation study.
>
> Concerning the lack of motivation for the problem, we attempted to address the motivation of a particular modeling approach taken for wind farms in lines 174 to 187. The two ablations that we found more valuable to report were focused on what separates this model from other models in the literature for the wind farm problem. These are summarized in tables 2 and 3.
>
> > ... need to be extremely solid on the experimental side ... in order to make up for the somewhat limited contributions in terms of novelty ... pavpu or etc...
>
> We indeed do not claim to here present a contribution along the lines of discovery of VAEs or Graph Networks. The claimed novel part of this work can be clearly seen in table 1. We find that this is of interest to the community, at least as we have come to perceive from the statements of the further reviewers.
>
> > There are multiple notational mistakes. ...
>
> 1. this is a typographical error, which we will correct.
> 2. We use this notation because the $G_h$ is a known parameter (the same reason we denote it with blue color in figure 1). This is not an error.
> 3. (see 4)
> 4. we thank the reviewer you for this observation. The $D_{KL}(q(z)||\cdot)$ operator denotes $\int_z q(z) \log \frac{q(z)}{\cdot} dz$ so **there is no error**. The KL divergence *is* an expectation over “$\mathbf{z}$”, that is why “$\mathbf{z}$” does not have an index.
>
> > If the model is Bayesian Graph modeling, [1] ...
>
> We would like to point out that the literature is very rich and often diverging approaches to GraphML and representation of uncertainty. Many very interesting and valuable models do not fit in the view offered in this paper. This work does not aspire to be a review paper. After all, a review paper is probably not even warranted, given the rate at which research in Graph ML is expanding. We will, however, consider including these references.
>
> > Equation 6 assumes ...
>
> The existence of relations is generally assumed fixed in Graph Net models. This is a Gaussian over the distribution of the random features. However, reducing/adapting the density of the graph by using the ideas from the suggested references [1] and [2] may yield interesting extensions of this framework. We let such investigations to potential future works.
>
> > I cannot follow lines 91 - 95 in that place. You might want to put them in the discussion or related works.
>
> Indeed some parts from these lines may better fit to the discussion. Thank you for the suggestion.
>
> > Validation could be used for early stopping. It is not clear to me how you used test data for this.
>
> For model comparison, it is enough to have a two-way split in train/validation (or train/testing split if you wish). If this comment pertains to the validity of the presented results, since the best performing models had not early-stopped for the results presented, there should be no doubt as to whether early stopping on a validation set positively affected the accuracy of the presented results.
>
> > It is not clear how the authors change the name of the model from RVAE to CRVAE and what are the differences.
>
> Perhaps re-naming the model Conditional RVAE (CRVAE) would be warranted. The facilitation of conditioning of nodes, edges, and global variables, for both the encoder and decoder, is one of the features that allow the unification of the other models cited and the connection with NPs and the [Conditional VAEs with arbitrary conditioning](https://arxiv.org/pdf/2008.09469.pdf) applied to graph ML.
>
> > *Limitations And Societal Impact:*
> > The complexity of the model and dealing with large graphs can also be considered as the limitation of the model.
>
> Dealing with large graphs is indeed a limitation of the proposed model and we will add a relevant comment. However, we believe that the view offered in our work still allows for straightforward adaptations to more efficient message passing.
>
> ## Closing
> We thank the reviewer for the detailed review. We believe we addressed the main critical comments (and in particular the notation mistakes, which were a typographical mistake (missing $=$) and a *slight* misunderstanding from the side of the reviewer on $D_{KL}$.

---

> > ### Comment · Reviewer_2qz2 · 2021-08-18
> > **Concern about the NP claim**
> >
> > I have carefully read the rebuttal and the paper again. While it was helped me to address some parts of my previous concerns, it raises some new concerns that are critical.
> >
> > First, I believe the paper only proposes a generalized formulation of a probabilistic model for graph-structured data that incorporates graph networks (I need to update my initial summary). However, I believe the authors did not support their claim that the framework is general enough to be Neural Processes (NP). Here are my concerns:
> >
> > While I'm an expert on Bayesian learning and graph neural networks, I still believe the paper is hard to follow. Let me go through the details in two different scenarios:
> >
> > 1) Initially, I thought the model is learning a single continuous latent space for multiple observed graphs, as in line 73 and above their main equation (i.e. eq 4), they assumed "multiple graph observations $G_x^{(1)}, \dots, G_x^{(i)}$". In such a case, the model could learn a single $G_z$ for all observations, similar to NPs in non-structured data. However, based on the rebuttal I found this is not the case.
> >
> > 2) The authors claim that they plan to combine local and global latent space, similar to DSNPV and ATTNNP. However, they did not support the claim in an acceptable way. Based on the author's formulation, they are referring to node (and edge) representation as local latent space, which is correct and similar to other VAEs on the graph. The problem came in the situation where the authors are constructing their model based on the latent space $\mathbf{u}_z$ as a global latent space for all the nodes, and conclude that they could reach out to an NP.  The global latent space does not necessarily lead to a stochastic process, rather there are two necessary conditions that have to be satisfied during the construction of such a model: exchangeability and consistency [1-3]. Without that, I cannot see any other way to show the proposed global latent space is leading to a stochastic process. This should not be an easy problem in this setting:
> >
> > - Joint distribution of all nodes needs to be consistent and permutation invariant, and the authors did not provide any proof for this.
> > - Even $\mathbf{u}_z$ is not global in terms of multiple graph observations.
> > - I disagree with the authors, that conditioning on the context in the decoder will lead to having an NP. For example, one can use the context point in the decoder in CVAE as well, which is not an NP.
> >
> > In this case, the novelty of this paper in terms of modeling is just adding another latent space, i.e. edge and their attributes to other graph latent space modelings.
> >
> > Apart from these modeling issues, it is not still clear to me why the probabilities are not conditioned on the context and the authors are using semi-columns. Do you construct it from the input or not?
> >
> > Still, it is not clear to me what kind of uncertainties the author is talking about. In VAE and CVAE settings, we have an uncertainty over the posterior of each sample/node. In NPs, we have uncertainty over the function. It would be useful to plot samples of curves conditioned on an increasing number of context points similar to Figure 1 in [1].
> >
> >
> > Even in the ablation study and the comparison with GP, it is not clear which part of the model provides the benefit. With lower ELBO, can we really say that the inductive biases are better encoded than GP?
> >
> > In line 272, the authors state "keep the conditional RVAE model closer to the NP formulation, the approximate posterior [...] is used also for the learned prior". The problem in the NPs, is the intractability of the model to the prior which forces to use posterior. If this is not the case, why one needs to use the posterior to construct the prior?
> >
> >
> > [1] Garnelo, Marta, et al. "Neural processes." arXiv preprint arXiv:1807.01622 (2018).
> >
> > [2] Louizos, Christos, et al. "The functional neural process." arXiv preprint arXiv:1906.08324 (2019).
> >
> > [3] Wang, Qi, and Herke Van Hoof. "Doubly Stochastic Variational Inference for Neural Processes with Hierarchical Latent Variables." ICML 2020.

---

> > > ### Author Response · Authors · 2021-09-07
> > > **Offering further explanations to support the claim that (1) NPs are Conditional RVAEs with DeepSet encoder, a global latent and a special decoder (2) RVAE satisfies node/edge permutations (and not exchangeability in the general case)**
> > >
> > > We thank the reviewer for taking the time to offer a second review. We address the points of the reviewer in what follows:
> > >
> > > > First, I believe the paper only proposes **a generalized formulation of a probabilistic model for graph-structured data that incorporates graph networks**
> > >
> > > The updated description of what the paper offers is more accurate. More specifically, the paper introduces a conditional VAE model for graph-structured data.
> > >
> > > > However, I believe the authors did not support their claim that the framework is **general enough** to be Neural Processes (NP).
> > >
> > > Please refer to figure 1-c in our paper (adaptation of the RVAE to a NP) and consider the correspondence we point to between the computational parts of the RVAE-like adaptation and figure 1-b in the NP paper. The figure illustrates the correspondence between computational graphs between the two schemes, which can also be seen in the notebook `RVAE_NP_1DRegression.ipynb` in the supplemental material. We argue that the correspondence between the computational graph and the loss function (specialized for NP) are correct. Therefore the RVAE can be adapted to a NP (fig1-c). From the reviewers' comments, it seems that further elaboration on how the RVAE model adapts to a NP model is needed.
> > >
> > > >Here are my concerns:
> > > > 1. Initially, I thought the model is learning a single continuous latent space for multiple observed graphs, as in line 73 and above their main equation (i.e. eq 4), they assumed "multiple graph observations ". In such a case, the model could learn a single G_z for all observations, similar to NPs in non-structured data. However, based on the rebuttal I found this is not the case.
> > >
> > > Here perhaps the reviewer meant $\mathbf{u_z}$ (a $G_z$ with only global variables). The full model features latents for nodes and edges (table 1 and figure 1-a).
> > >
> > > In the NP adaptation (described in lines 138-141) since a DeepSet is used, only a global latent is meaningful. Please also note that the NP model does not learn a single $G_z$, since the loss is not a function of $G_z$ (eq 4). The $p(z)$ in the NP model is "dataset dependent" only in the absence of context points. In the RVAE modeling framework, this becomes $p(G_z; G_h)$ since the $G_h$ contains graph-structured parameters we already know. Hopefully, this clarifies this point.
> > >
> > > > 2. The authors claim that they plan to combine local and global latent space, similar to DSNPV and ATTNNP. However, they did not support the claim in an acceptable way.
> > >
> > > We attempt a clarification referring to the figures offered in the original DSVNP paper.
> > > Firstly, the "local" variable in DSVNP corresponds to the "node latent variable" in our model.
> > >
> > > Consider figure 1-d in the DSVNP paper:
> > >  * The blue arrows consist of a node function and a node-to-global aggregation (to yield $z_G$ or $\mathbf{u_z}$ in our model).
> > >  * Blue arrows correspond to node-to-global blocks of the encoder in an RVAE. No message passing is performed (no edges - the DSVNP encodes absolute node $((x_C, x_*), (y_C, y_*))$ information).
> > >  * At test time, in DSVNP the decoder receives $z_G$, $z_*$ and $x_*$. This corresponds to an RVAE decoder with no message passing, and a node function that receives the "global latent" (or \mathbf{u_z}) $z_G$ and the "local latent" (or node latent $V_{\mathbf{z}}$) $z_*$. The red arrow from $z_G$ to $z_*$ is **not supported** by the RVAE (which uses the GN computation block) but can be straightforwardly included with a hierarchical extension (as claimed in lines 91-93 in the paper). The remaining red arrows are equivalent to what corresponds to "node functions" in our model (operating independently on nodes).
> > >
> > > From figure 1-c of the DSVNP paper, similar correspondences can be drawn for the AttNP work. Also, by noting the well-established correspondence between GNs on fully connected graphs and transformers, one can draw straightforward analogies between our work and AttNPs. We hope this further clarifies these claimed connections.
> > >
> > > > The global latent space does not necessarily lead to a stochastic process, ... that conditioning on the context in the decoder will lead to having an NP.
> > >
> > > We refer the reviewer to the AttNP paper, the last paragraph before section 2.2, where similarly to the RVAE, it is clearly stated that indeed consistency is sacrificed (at least theoretically).
> > > However, note that permutation invariance (on edge and node index permutation) **is** satisfied due to the use of GNs. We would like to note, however, that our claim is that the proposed framework is on incorporation of VAEs in continuous graph-structured data, not necessarily stochastic processes!
> > >
> > > > I disagree with the authors, that conditioning on the context in the decoder will lead to having an NP. For example, one can use the context point in the decoder in CVAE as well, which is not an NP.
> > >
> > > We further clarify the point related to the claim in lines 136-141 (conditioning the decoder of a VAEAC appropriately gives a NP). Conditioning *appropriately* on an embedding of the context, as shown in fig1-c for the $V_h$ (that contains $x$ for the 1D GP example) yields the NP model. We will rephrase this part to clearly reflect the point that we are not referring to *any* conditioning. Moreover, we will - for clarity - include training time and testing time computational graphs in the supporting materials.
> > >
> > > > In this case, the novelty of this paper in terms of modeling is just adding another latent space, i.e. edge and their attributes to other graph latent space modelings.
> > >
> > > We also claim that the view offered is valuable towards unifying NPs and GN-based VAEs (fig 1). We found the proposed approach quite flexible since it allows for building translation equivariant NP-like models and improves VAE performance by including the underlying geometric structure, which are the examples included in the paper.
> > >
> > > > it is not still clear to me why the probabilities are not conditioned on the context
> > >
> > > As mentioned above, it is possible to condition only on $G_h$ (and empty context), i.e., known graph-structured variables. Then, a data-dependent (graph) prior is used to sample $G_z$.Perhaps this comment further clarifies the reason for using a semicolon $p(G_x; G_h )$ instead of $p(G_x| G_h)$. It becomes unclear that $G_h$ is of a different "type" when writing $p(G^T_x|G^{C}_x , G_h)$ and $p(G^T_x|G^C_x ; G_h)$ is preferred.
> > >
> > > > ... the authors are using semi-columns. Do you construct it from the input or not?
> > >
> > > $G_h$ variables are parameters and not random variables (e.g., encoding the geometry of the farm). The model does not aim at inferring anything about the $G_h$ variables - they are inputs (not reconstructed in the VAE output). In the 1D GP example, for instance, the nodes $V_x$ of $G_x$ contain instances of the random variable $y$ and the nodes $V_h$ of $G_h$ contain the *indexing* variable $x$ (fig 1). A concatenation of these node arguments is then passed to the GN encoder or the DeepSet encoder (for the NP case).
> > >
> > > > Still, it is not clear to me what kind of uncertainties the author is talking about. In VAE and CVAE settings, we have an uncertainty over the posterior of each sample/node. In NPs, we have uncertainty over the function.
> > >
> > > Since the $G_z$ contains a node, an edge, and a global latent, which enter the respective GN functions, you may consider the graph latent as parametrizing a posterior over GraphNet functions conditioned on observed data $G^{C}_x$ (and/or observed conditioning data $G_h$ - e.g., the farm geometry encoded in edge features).
> > >
> > > >It would be useful to plot samples of curves conditioned on an increasing number of context points similar to Figure 1 in [1].
> > >
> > > Due to space limitations, one such plot is included in the supporting materials (together with the code to reproduce it in a notebook).
> > >
> > > > Even in the ablation study and the comparison with GP, it is not clear which part of the model provides the benefit. With lower ELBO, can we really say that the inductive biases are better encoded than GP?
> > >
> > > In the NP comparison, please consider the last column of table 4. Since it is possible to encode relational information in the edge features, the model generalizes beyond $x_T \in [0,1]$. The goal of neural models for distribution approximation (such as the claimed RVAE model) is to circumvent the definition of the kernel that would be necessary for a GP. We demonstrate that the RVAE offers a flexible way to encode relational inductive biases for neural models for conditional distributions. Namely, the known relations are encoded in the conditioning graph data $G_h$.
> > >
> > > >In line 272, the authors state "keep the conditional RVAE model closer to the NP formulation...why one needs to use the posterior to construct the prior?
> > >
> > > Take note of the VAEAC implementation for imputation problems. A $q_\phi$ network (the proposal network) that contains the masking and the full input and a separate $p_\psi$ network (the prior network) are used. This sentence is meant to explain that the *same network* is used for the respective $q_\phi$ and $p_\psi$ (not that some posterior is used for a prior).
> > >
> > > ## Closing
> > > We claim the permutation invariance of the proposed model is inherited from GNs and that the most general form of the model is a VAE (as clearly indicated also from the naming) and not a NP. We pointed to additional specific points in the DSVNP paper, where we believe the connection to the proposed modeling framework becomes even clearer. We hope that the above discussion, which explicitly addresses the raised points of criticism, offers a clearer view of the paper. From the comments of the reviewer, an elaboration offering a clearer path from the Relational VAE to NP is required. However, we would like again to stress the correspondence between the computational graphs in fig1-c and the implementation in the code provided in the supplemental material (`RVAE_NP_1DRegression.ipynb`).

---

> > > > ### Comment · Reviewer_2qz2 · 2021-09-08
> > > > **I still believe the author's responses has not convincing**
> > > >
> > > > Thanks to the author for the response, however, I would like to point out that you have to consider the timeline of the rebuttal. I have raised my concern on Aug 18, however, the authors answered so late on Sep 7, after the discussion period has been closed.
> > > >
> > > > But to help the authors to prepare the submission for the next avenue, I would like to point out to main points in their response:
> > > >
> > > > **Consistency:** I agree with the author that AttNP does not satisfy consistency in the
> > > > contexts which is due to considering both local and global latent variables modeling. However maximum-likelihood learning can be interpreted as minimizing the KL between the consistent conditional distributions of the data-generating stochastic process and the corresponding conditional distributions of the NP. Hence the NP can be seen as approximating
> > > > the conditionals of the consistent data-generating stochastic process. Here, I cannot see similar property. Therefore, the authors need to somehow show this. I believe block diagrams are not enough to show a model is NP or not. Please check Functional Neural Processes to see why theoretical proof is needed here.
> > > >
> > > > **Permutation invariance:** Please note that the GNNs are permutation invariance in terms of nodes and edges, however, this is not enough in terms of defining a global latent space as an NP. In the graph, the authors need to show that the model is permutation invariance in terms of attributes in addition to the nodes and edges that are already satisfied by the GNs. In general, this is more like time-series modeling.

---

### Official Review · Reviewer_4JjX · 2021-07-19

**Rating:** 6
**Confidence:** 3

**Summary:**

The authors propose to extend Graph Nets (GNs) by allowing edge-bound uncertainty and edge-level conditioning using a Variational Bayes formulation.

**Limitations And Societal Impact:**

The authors have very briefly addressed the limitations and potential negative societal impact of their work.

**Main Review:**

The paper is clear although (necessarily) not self-contained and it therefore becomes at times hard to follow for someone who is not deep into the subject matter.  The authors should perhaps try to better explain the flexibility and the generalisation capacity of the model proposed: would the model break or suffer catastrophic failures if the nodes are not placed in a quasi regular lattice? The reader would assume that as the turbines placement becomes more irregular, it becomes more difficult to leverage and transfer information from various parts of the network, but does the edge modelling give more robustness in these cases? Quantifying these notions via empirical experimentation would  improve the paper impact.
Some empirical comparison with Gaussian Processes for the turbine application case is missing.
Further improvements to the paper could include a more detailed discussion on the implications and limitations of factorizing the graph priors by the edges, nodes and global variables separately.
Finally, perhaps a more representative artificial case could be devised to showcase the improved capacity to model relational information since in the proposed example the edges simply encode the distance between nodes and this is a type of information that can be readily derived given that the node attributes include the node coordinates, i.e. it is not something that is accessible only through modelling an relational process that is independent from the node process.

###After authors' reply
Many thanks to the authors for the effort in replying all our concerns. After reviewing the comments raised by the other reviewers I find myself sharing several of their concerns, especially the need of a much improved clarity in explaining the relationship of their proposal to Neural Processes. I maintain that for non-expert readers it would be beneficial to be presented with intuitive artificial cases where the relational nature of the objective is clear and unquestionable.


**Time Spent Reviewing:**

3

---

> ### Author Response · Authors · 2021-08-10
> **Response to reviewer**
>
> We would like to thank the reviewer for the interesting suggestions and for suggesting the acceptance of the submission. Point-by-point answers to the suggestions and issues raised to follow.
>
> > *  would the model break or suffer catastrophic failures if the nodes are not placed in a quasi regular lattice?
>
> > * The reader would assume that as the turbines placement becomes more irregular, it becomes more difficult to leverage and transfer information from various parts of the network, ...
>
> This comment allows us to expose an interesting property of GNs, relating to generalization. The generalization of the model is best assessed by separately considering edge, node, and global functions.
> Given the geometry of a farm, there is a distribution of the relative orientation, with respect to north, of the vector connecting two turbines, defined by parameters $[\cos(\phi_{ij}), \sin(\phi_{ij})]$, as well as of the relative distance parameter between the turbines, $d_{ij}$, in a farm (see figure 3 of the supplemental materials). The static edge variables are
>
> $$
> \\mathcal{E}_h = [\cos(\\phi), \\sin(\\phi), d]
> $$
>
> for $i,j$ any pair of turbines connected in the graph (including self-edges). (Due to rendering issues, $ij$ removed from the equation above)
>
> Since the edge functions are shared between the turbines, the learning signal for the GN edge function contains the directional dependence from all pairs of turbines. It is expected that the model will generalize well for other farms that comprise parameters, $d_{ij}$ and $\phi_{ij}$, which do not significantly deviate from the distribution that characterizes the training farms.
>
> The generalization capacity, thus, depends on the range of parameters used in training. If the model were trained on a regular lattice, it would generalize well *only* to further regular lattice geometries. If, instead, it is trained on a diverse set of relative $\phi_{ij}$ angles (which - by design - is a feature of the training farm used in the simulated example) the model can generalize over a broader geometrical configuration.
>
>
> > but does the edge modelling give more robustness in these cases? Quantifying these notions via empirical experimentation would improve the paper impact.
>
> Indeed this would be an interesting exploration. We thank the reviewer for the suggestion and we will consider it for future studies.
>
> >Some empirical comparison with Gaussian Processes for the turbine application case is missing.
>
> Indeed, a properly calibrated approximate sparse GP approach would be interesting. However, a part of the initiating motivation for creating the RVAE was the high computational and memory costs of creating a dense GP for this problem. We do, though believe that principled approximate GPs may be useful, especially for the purpose of comparison, and this is left to potential future work.
>
>
> >Finally, perhaps **a more representative artificial case** could be devised to showcase the improved capacity to model **relational information** since in the proposed example the edges simply encode the distance between nodes and this is a type of information that can be readily derived given that the node attributes include the node coordinates, i.e. it is not something that is accessible only through modeling an relational process that is independent from the node process.
>
> Indeed, the synthetic 1D GP example does not expose all of the merits of the proposed approach. We opted to adopt this here precisely for its simplicity and usefulness as a toy example/demonstration, which facilitates the comparison over different employed algorithms. Note, however, that the conditional sampling from a Gaussian Process is *relational* and translation equivariant (the kernel takes $|x_i - x_j|$ as input). Therefore, one can get just as good results by completely neglecting "$x$" from the nodes and using only the edge features (namely $f(x_i, x_j) = e^{-c \cdot |x_i - x_j|} $). When also node features exist, the model essentially learns to ignore them, as evidenced by the good performance in the last column of Table 4.

---

### Decision · Program_Chairs · 2021-09-27

**Decision:**

Reject

**Comment:**

This submission considers a probabilistic model for graph-structured data that incorporates graph networks. It also attempts to connect/extend CVAEs, NPs for graph data. However, the reviewers remained unconvinced that the generalisation provides theoretical insights or practical benefits. In addition, all reviewers agreed the exposition of the work is hard to follow and the experiment section requires more work. Thus, the paper in its current form is not ready to be accepted.

---

> ### Public Comment · ~Charilaos_Mylonas1 · 2021-11-23
> **Original response to review decision**
>
> We would like to thank the committee for the time they have spent reviewing our manuscript.
>
> In summary, the main issues raised and attempted to be addressed are as follows:
>
> 1. Comments about **notation mistakes** and **mistakes in derivations** (in particular the formula for the KL divergence) pertained to miscomprehensions from the reviewers' side and one typographical mistake.
> 2. Comments about the proposed approach for modeling distributions not being general enough to represent a stochastic process were addressed by pointing out that a stochastic process is a particular case of a joint distribution satisfying specific properties and that the proposed model focuses on distributions over graph-structured data. Unfortunately, we have not managed to convince the reviewers on the correspondence between Figure 1-c in our paper and Figure 1-b in [(Garnelo, 2018)](https://arxiv.org/pdf/1807.01622.pdf) and that the NP model is derived with a DeepSet encoder (operating on sets with no edges) and a decoder that corresponds to the "node-block" of a Graph Network that accepts only global and (target) node inputs.
> 3. Points raised on clarity of exposition were addressed by providing further clarifications and more explicit connections to specific parts of cited papers in the literature (see [this official](https://openreview.net/forum?id=mrsx7ninrtU&noteId=76AWapgqDf) response comment offered on Sept. 7) and pointing to the provided code and the computational diagrams of the RVAE and the NP in Figure 1.
> 4. The point on further experiments for supporting the usefulness of the proposed framework was addressed by pointing out that the cited models were benchmarked on a large variety of tasks. We have presented comparisons only to the NP model for the wind farm dataset and a 1D regression example, showing that encoding relative position in edges allows for equivariant models. We would like to also note that models in the literature have different overlapping properties but sometimes clearly non-overlapping merits. We endorse a purposeful selection of different features of the available models instead of the quest for a generally superior model. For instance, the absence of edge features in CNPs and NPs does not allow for modeling the relative orientation and positioning information necessary for capturing the effects observed in a wind farm. The absence of explicit parametrization for edge features in AttCNPs makes it hard to incorporate known relational/geometric information like the relative angles of nacelles of turbines, although we are aware that remedies to incorporating known edge features ($\mathcal{E}^h$ in our notation) do exist. On the other hand, the use of a single global latent in CNPs/NPs makes the computational requirements of these models scale linearly on context and target set size, which is a property not shared with AttCNP. Therefore our numerical experiments focused on demonstrating the advantages of the main technical novelty of our work which is an approach for learning distributions for general graph-structured data, and contrast our approach to a "set" function approach supported by NPs. We show that this approach allows for generalizing generative models for wind farms in unseen farm layouts and constructing equivariant NP models (by employing a special choice of equivariant edge features $\mathcal{E}^h$) for the 1D regression problem (as in ConvCNPs).
>
> For the benefit of future readers, in [this comment](https://openreview.net/forum?id=mrsx7ninrtU&noteId=yR1NhRrCGs) it is stated that what we present **" is more like time-series modeling"**. We would like to point out that we do not share this perspective.
>
> Finally, a point was raised by **2qz2** on the timing of our second response that we would like to address: Reviewer **zzT8** stated that he/she was influenced by point (2) raised by reviewer **2qz2**, who added an official comment on Sept. 5. This is why our response to the second comment of **2qz2** was on Sept. 7. Reviewer **4JjX** also lowered the score while citing discussions with the other reviewers but updated the review on Sept. 1 (1 day before the end of the discussion period). Although clearly, we have not managed to convince reviewer **2qz2**, we claim that point 2 was sufficiently addressed by referring to the provided code and specific parts of the cited literature. Moreover, our comment on [Sept. 7](https://openreview.net/forum?id=mrsx7ninrtU&noteId=76AWapgqDf) may partially address the point on clarity.
>
> We would like to point future readers to [our comment](https://openreview.net/forum?id=mrsx7ninrtU&noteId=RymY9amtBeG) in response to reviewer **zzT8**, which offers a clarification on the graph prior and how it is conditioned on the known graph features.
>
> In our reading of the reviews, the reviewers seem to agree to one of the main claimed merits of this paper: it is a general framework for deep latent variable models for continuous-valued graph-structured data.
>
> We would like to close by expressing our gratitude to the reviewers for their time and effort. We highly value their constructive comments on improving the clarity of the exposition and making the claimed connections more explicit, as well as on demonstrating the proposed general model in other datasets, which we will adopt in the future.